# Hypoxia-induced metabolic stress in retinal pigment epithelial cells is sufficient to induce photoreceptor degeneration

Toshihide Kurihara[1†‡], Peter D Westenskow[1,2†], Marin L Gantner[2], Yoshihiko Usui[1§], Andrew Schultz[3], Stephen Bravo[1], Edith Aguilar[1], Carli Wittgrove[1], Mollie SH Friedlander[1], Liliana P Paris[1], Emily Chew[4], Gary Siuzdak[3], Martin Friedlander[1*]

[1]Department of Cell and Molecular Biology, The Scripps Research Institute, La Jolla, United States; [2]The Lowy Medical Research Institute, La Jolla, United States; [3]Center for Metabolomics, The Scripps Research Institute, La Jolla, United States; [4]National Eye Institute, National Institutes of Health, Bethesda, United States

*For correspondence: friedlan@scripps.edu

[†]These authors contributed equally to this work

Present address: [‡]Department of Ophthalmology, Keio University School of Medicine, Tokyo, Japan; [§]Department of Ophthalmology, Tokyo Medical University Hospital, Shinjuku, Tokyo, Japan

Competing interests: The authors declare that no competing interests exist.

**Abstract** Photoreceptors are the most numerous and metabolically demanding cells in the retina. Their primary nutrient source is the choriocapillaris, and both the choriocapillaris and photoreceptors require trophic and functional support from retinal pigment epithelium (RPE) cells. Defects in RPE, photoreceptors, and the choriocapillaris are characteristic of age-related macular degeneration (AMD), a common vision-threatening disease. RPE dysfunction or death is a primary event in AMD, but the combination(s) of cellular stresses that affect the function and survival of RPE are incompletely understood. Here, using mouse models in which hypoxia can be genetically triggered in RPE, we show that hypoxia-induced metabolic stress alone leads to photoreceptor atrophy. Glucose and lipid metabolism are radically altered in hypoxic RPE cells; these changes impact nutrient availability for the sensory retina and promote progressive photoreceptor degeneration. Understanding the molecular pathways that control these responses may provide important clues about AMD pathogenesis and inform future therapies.

## Introduction

Uninterrupted blood flow and an intricate and architecturally optimized network of photoreceptors, interneurons, glia, and epithelial cells are required for vision. The primary blood supply for photoreceptors is the choriocapillaris, an extraretinal fenestrated capillary bed. A layer of extracellular matrix proteins, Bruch's membrane, lies adjacent to the choriocapillaris, and a monolayer of retinal pigment epithelium (RPE) cells divides Bruch's membrane from the photoreceptors. The choriocapillaris, Bruch's membrane, RPE, and photoreceptors function as one unit, with the choriocapillaris providing fuel for phototransduction, and Bruch's membrane and RPE cells filtering and regulating the reciprocal exchange of oxygen, nutrients, biomolecules, and metabolic waste products between the circulation and retina. RPE also provide critical support for both photoreceptors and the choriocapillaris (*Strauss, 2005*) in large part by generating vascular endothelial growth factor (VEGF), a cytokine required for choriocapillaris development and maintenance (*Kurihara et al., 2012*; *Le et al., 2010*; *Marneros et al., 2005*; *Saint-Geniez et al., 2009*). Defects in this unit, including reduced choriocapillaris density, the presence of sub-RPE lipid-rich deposits, and RPE/photoreceptor dysfunction, are characteristic of age-related macular degeneration (AMD), a common vision-threatening disease whose prevalence is steadily increasing globally (*Friedman et al., 2004*; *Wong et al., 2014*). Several genetic and lifestyle risk factors have been identified but no cure exists (*Bird, 2010*).

**eLife digest** Cells use a sugar called glucose as fuel to provide energy for many essential processes. The light-sensing cells in the eye, known as photoreceptors, need tremendous amounts of glucose, which they receive from the blood with the help of neighboring cells called retinal pigment epithelium (RPE) cells. Without a reliable supply of this sugar, the photoreceptors die and vision is lost.

As we age, we are at greater risk of vision loss because RPE cells become less efficient at transporting glucose and our blood vessels shrink so that the photoreceptors may become starved of glucose. To prevent age-related vision loss, we need new strategies to keep blood vessels and RPE cells healthy. However, it was not clear exactly how RPE cells supply photoreceptors with glucose, and what happens when blood supplies are reduced.

To address this question, Kurihara, Westenskow et al. used genetically modified mice to investigate how cells in the eye respond to starvation. The experiments show that when nutrients are scarce the RPE cells essentially panic, radically change their diet, and become greedy. That is to say that they double in size and begin burning fuel faster while also stockpiling extra sugar and fat for later use. In turn, the photoreceptors don't get the energy they need and so they slowly stop working and die.

Kurihara, Westenskow et al. also show that there is a rapid change in the way in which sugar and fat are processed in the eye during starvation. Learning how to prevent these changes in patients with age-related vision loss could protect their photoreceptors from starvation and death. The next step following on from this research is to design drugs to improve the supply of glucose and nutrients to the photoreceptors by repairing aging blood vessels and/or preventing RPE cells from stockpiling glucose for themselves.

While current evidence suggests that AMD is a spectrum of closely related multifactorial polygenic diseases (*Bird et al., 2014*), 10 year clinic-based data from the Age-Related Macular Degeneration Study (AREDS; n = 4757) showed that the major risk factors include aging, severity of drusen (sub-RPE deposits), and RPE abnormalities (*Chew et al., 2014*). Early AMD is characterized by pigmentary changes and appearance of drusen (*Gass, 1973*; *Pauleikhoff et al., 1990*; *Sarks, 1976*; *Wang et al., 2010*). In most cases early AMD proceeds towards geographic atrophy (or 'dry' AMD), a condition defined by focal photoreceptor, RPE, and choriocapillaris loss and thickening of Bruch's membrane with immunomodulatory proteins and lipids (*Bird, 2010*; *Jager et al., 2008*; *Zarbin, 1998*). While the primary defect could occur in Bruch's membrane (*Pauleikhoff et al., 1990*; *Bressler et al., 1990*; *Mullins et al., 2011*), the choriocapillaris (*Ramrattan et al., 1994*; *Spraul et al., 1996*; *Spraul et al., 1999*), or photoreceptors (*Sarks, 1976*; *Hogan, 1972*; *Sarks et al., 1988*), most evidence suggests that it probably occurs in RPE cells. Granules enriched with lipid-rich residues, lipofuscin, accumulate normally in aging RPE cells (*Bazan et al., 1990*), but abnormal accretions are observed in patients with geographic atrophy in a band directly surrounding the lesion (*Feeney-Burns et al., 1984*; *Holz et al., 1999*; *von Ruckmann et al., 1997*). Lipofuscin renders the RPE more sensitive to blue light-induced damage (*Rozanowska et al., 1995*; *Schutt et al., 2000*; *Sparrow et al., 2000*), impairs RPE functions (*Holz et al., 1999*; *Finnemann et al., 2002*; *Lakkaraju et al., 2007*; *Sparrow et al., 1999*), and is potentially toxic for RPE cells (*Schutt et al., 2000*). The downstream effects of RPE loss are catastrophic, and result in choriocapillaris attenuation and photoreceptor degeneration in late stage AMD patients (*Bhutto and Lutty, 2012*; *Coscas et al., 2014*; *Jonas et al., 2014*; *Lee et al., 2013*; *Sohrab et al., 2012*; *McLeod et al., 2009*).

Assuming, therefore, that the critical event of AMD pathogenesis occurs in RPE cells, how can the onset of the other early clinical manifestations of the disease in neighboring cells and structures be explained? There is a growing body of evidence that choroidal and retinal blood flow is reduced in AMD (*Boltz et al., 2010*; *Remsch et al., 2000*). We hypothesize that hypoxia, a natural consequence of aging microenvironments (that is exacerbated by obesity and smoking) (*Blasiak et al., 2014*; *Chiu and Taylor, 2011*; *Morgado et al., 1994*; *Sagone et al., 1973*), in RPE cells may be a central

AMD risk factor based on the following lines of evidence: (a) RPE provide critical vasculotrophic support required for photoreceptor function (*Kurihara et al., 2012*); (b) Hypoxia alters lipid handling in other cell-types (*Glunde et al., 2008*; *Santos et al., 2012*; *Semenza, 2009*); (c) at least 40% of lipids in drusen are secreted by RPE (*Cao et al., 2013*); (d) the RPE secretome is sensitive to stress (*Wang et al., 2010*); and (e) in drusen rich zones of AMD patient eyes, vascular density is significantly reduced (*Mullins et al., 2011*). Therefore, hypoxia-mediated changes to the RPE lipidome and secretome could enhance lipofuscin accumulation and induce Bruch's Membrane lipidization and thickening, thereby exacerbating RPE dysfunction, choriocapillaris drop-out, and photoreceptor dysfunction. This vicious cycle of events could accelerate progression of AMD.

Hypoxia-inducible factor alpha subunits (HIF-αs) are the key transcription factors that mediate responses to hypoxia. Under normal conditions HIF-αs are constitutively expressed and targeted by von Hippel-Lindau protein (VHL) for ubiquitination and proteasomal degradation. VHL is inactivated at low oxygen tensions; this allows HIF-αs to translocate to the nucleus and activate a host of angiogenesis, glucose metabolism, erythropoiesis, and inflammation genes (*Semenza, 2011*). In this study we directly or indirectly hyperactivated HIF-αs in RPE by genetically perturbing components of the VHL/HIF/VEGF pathway using inducible and conditional gene ablation techniques. These manipulations altered lipid handling and glucose consumption of RPE cells, induced gross morphometric changes in RPE, reduced nutrient availability for the sensory retina, and promoted progressive photoreceptor atrophy. Understanding the effects of hypoxia on RPE metabolism, and learning how to control these effects, may provide insights for developing novel therapeutic strategies to treat retinal degenerative diseases.

## Results

### Choriocapillaris attenuation induces hypoxia in RPE and promotes photoreceptor degeneration

Based on the hypothesis that choriocapillaris attenuation and prolonged hypoxia in RPE cells induces photoreceptor death/dysfunction, we set out to catalog the temporal and spatial manifestations of hypoxia in retinas of mice with severe choriocapillaris deficits (*VMD2-Cre;Vegfa^{fl/fl}*) and correlate these manifestations with any corresponding anatomical and functional changes in photoreceptors. Transgenic mice harboring human vitelliform macular dystrophy-2 promoter-directed cre (*VMD2-Cre*) (*Le et al., 2008*) were used to ablate *Vegfa*; severe choriocapillaris vasoconstriction is observed in adult *Vegfa*-cKO mice three days post induction (dpi) (*Kurihara et al., 2012*). The first signs of RPE hypoxia, including nuclear HIF immunoreactivity (*Figure 1A*), accumulation of the hypoxic probe pimonidazole in RPE (*Figure 1A&B*; white arrows), and activation of a known panel of hypoxia-inducible target genes, were observed six months post induction (mpi) in the *Vegfa* mutants (*Figure 1—figure supplement 1&2*). However, we cannot exclude the possibility that low-grade hypoxia may occur in RPE or other retinal cells earlier at subthreshold levels of detection. Hypoxia in RPE induced several defects including severely distended basal infoldings, accumulation of lipid droplets within RPE cells (*Figure 1C*; yellow arrows), RPE cell hypertrophy (*Figure 1D*), and dramatic and progressive thickening of Bruch's membrane beginning at nine months post induction (*Figure 1C* red arrows; *Figure 1E* pseudo-colored blue). At 11 months post induction we also detected pigmentary abnormalities in fundus images (*Figure 2A*) and thinning of the photoreceptor cell layer (*Figure 2B*; red line) characteristic of photoreceptor degeneration. While RPE defects took months to manifest, defects in cone-driven pathways occurred within seven days of *Vegfa* ablation (*Kurihara et al., 2012*) and do not recover by 11 months post induction (*Figure 2C and D*, photopic). Surprisingly, rod-driven pathway defects were not observed until 11 months post ablation (*Figure 2C and D*, scotopic), suggesting that, for reasons that are unclear, rod photoreceptors are less sensitive to oxygen and nutrient deprivation than cones are.

### RPE can induce choriocapillaris vasodilation

Theoretically, RPE should be able to respond to hypoxia and improve circulation in the choriocapillaris by increasing basal VEGF secretion. To induce hypoxia we maintained 3D cultures of primary human RPE in 3% oxygen in a controlled chamber and analyzed the molecular and metabolic changes. The HIF target gene *Vegfa* was upregulated after exposure to low oxygen (*Figure 3A*) or

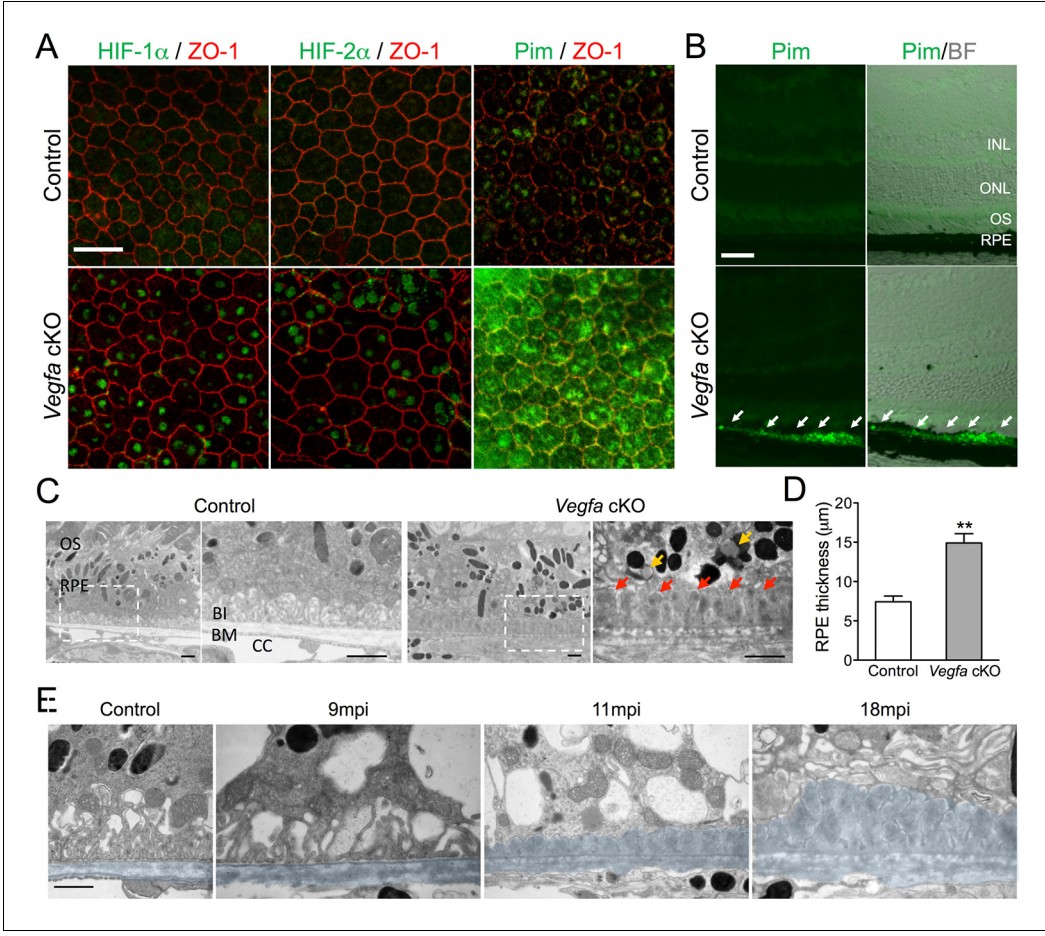

**Figure 1.** HIF-α accumulation precedes the induction of AMD-like features in *Vegfa*-cKO mice. (**A**) HIF-1α, HIF-2α, and pimonidazole (Pim; green) are detected 6 months post induction in flat-mounted RPE/choroids from *Vegfa* mutants but not in littermate controls. ZO-1 (red) labels the cell boundaries. (**B**) The hypoxia probe Pimonidazole (Pim; green) is detected specifically in the RPE (white arrows) of cross-sectioned *Vegfa* mutant retinas (probe labeling is shown alone (left) and overlaid over brightfield images (right) to emphasize the retinal anatomy). (**C**) Electron micrographs of littermate control (left) and *Vegfa*-cKO RPE 11 months post induction (right). Dashed squares in left panels are magnified in right panels. Note the absence of choriocapillaris in *Vegfa* mutants, accumulation of lipid droplets (yellow arrows) in the cytoplasm, and thickening of Bruch's membrane (red arrows). (**D**) Measured thickness values of the RPE of *Vegfa* mutant mice 11 months post induction (*n*=4) (see associated *Figure 1—source data 1*). (**E**) Electron micrographs of RPE/Bruch's membrane from control and *Vegfa*-cKO RPE taken 9, 11, and 18 months post induction. Note the progressive accumulation of material in Bruch's membrane (pseudo-colored light blue). Abbreviations: Pim=pimonidazole, INL=inner nuclear layer, ONL=outer nuclear layer (photoreceptor cell bodies), OS=photoreceptor outer segments, BI=basal infoldings of RPE cells, BM=Bruch's membrane, CC=choriocapillaris, mpi=months post [RPE-specific] induction. Scale bars=20 μm (**A**), 50 μm (**B**), 1 μm C&E.

The following source data and figure supplements are available for figure 1:

**Source data 1.** Source data for *Figure 1D*.

**Figure supplement 1.** Recombination efficiency in *VMD2-Cre* mice.

**Figure supplement 2.** Upregulation of HIF target genes in *Vegfa*-cKO RPE.

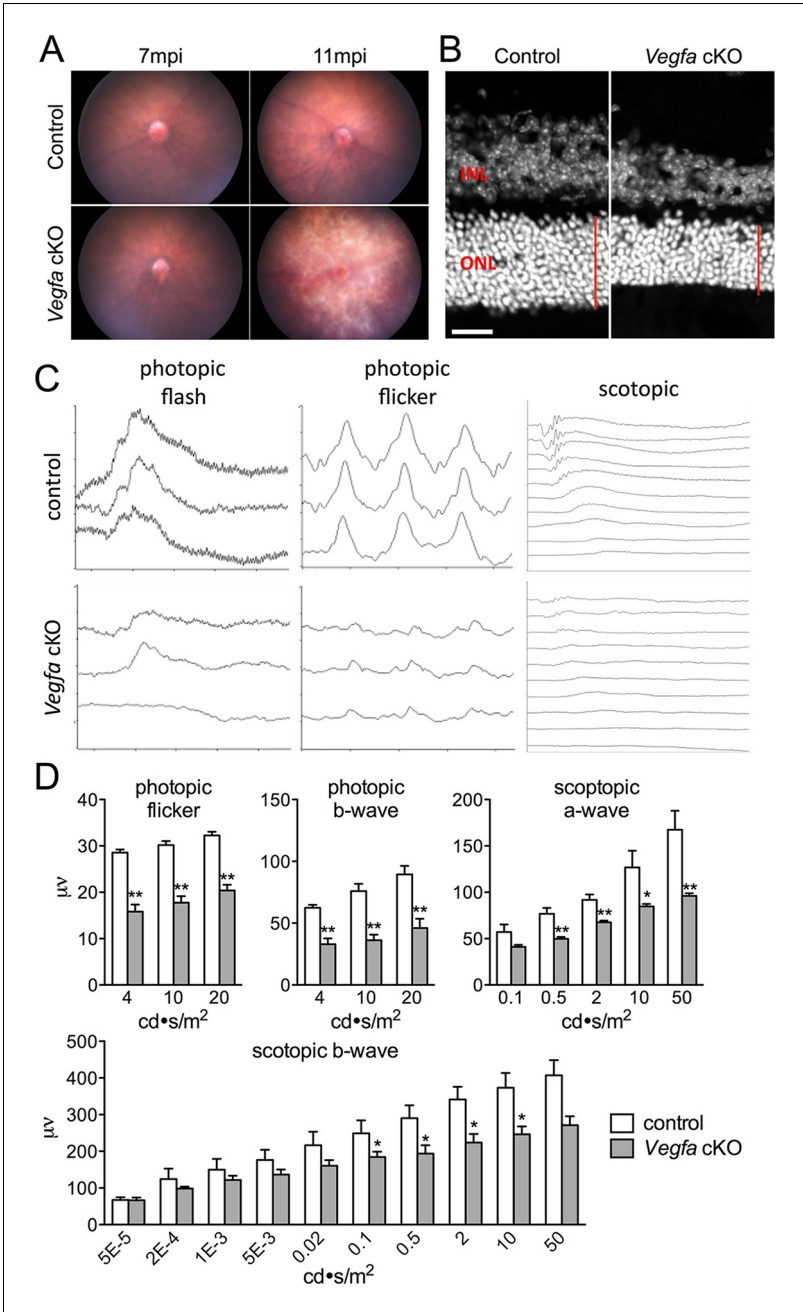

**Figure 2.** Photoreceptor atrophy and dysfunction is observed in late stage *Vegfa*-cKOs. (**A**) Color fundus images of littermate control and *Vegfa*-cKO mice 7 or 11 months post induction (mpi). No obvious changes are seen after 7 months, but significant changes indicative of retinal degeneration are seen 11 months post induction. (**B**) Photoreceptor atrophy, determined by observing thinning of the outer nuclear layer (ONL; red vertical line) is seen in DAPI labeled cryosectioned *Vegfa*-cKO retinas 18 months post induction 600 μm from the optic nerve head compared with controls. (**C**) Full-field ERGs performed on controls and *Vegfa*-cKO mice 11 months post induction (*n*=6) reveal rod dysfunction in both the a- and b-waves (scotopic), and cone dysfunction in the b-wave and flicker response (photopic flash & flicker) in *Vegfa*-cKO mice. (**D**) Quantification of ERGs (see associated *Figure 2—source data 1*). *p<0.05, **p<0.01. Error bars indicate mean plus s.d. Scale bar=20 μm.

The following source data is available for figure 2:

**Source data 1.** Source data for *Figure 2D*.

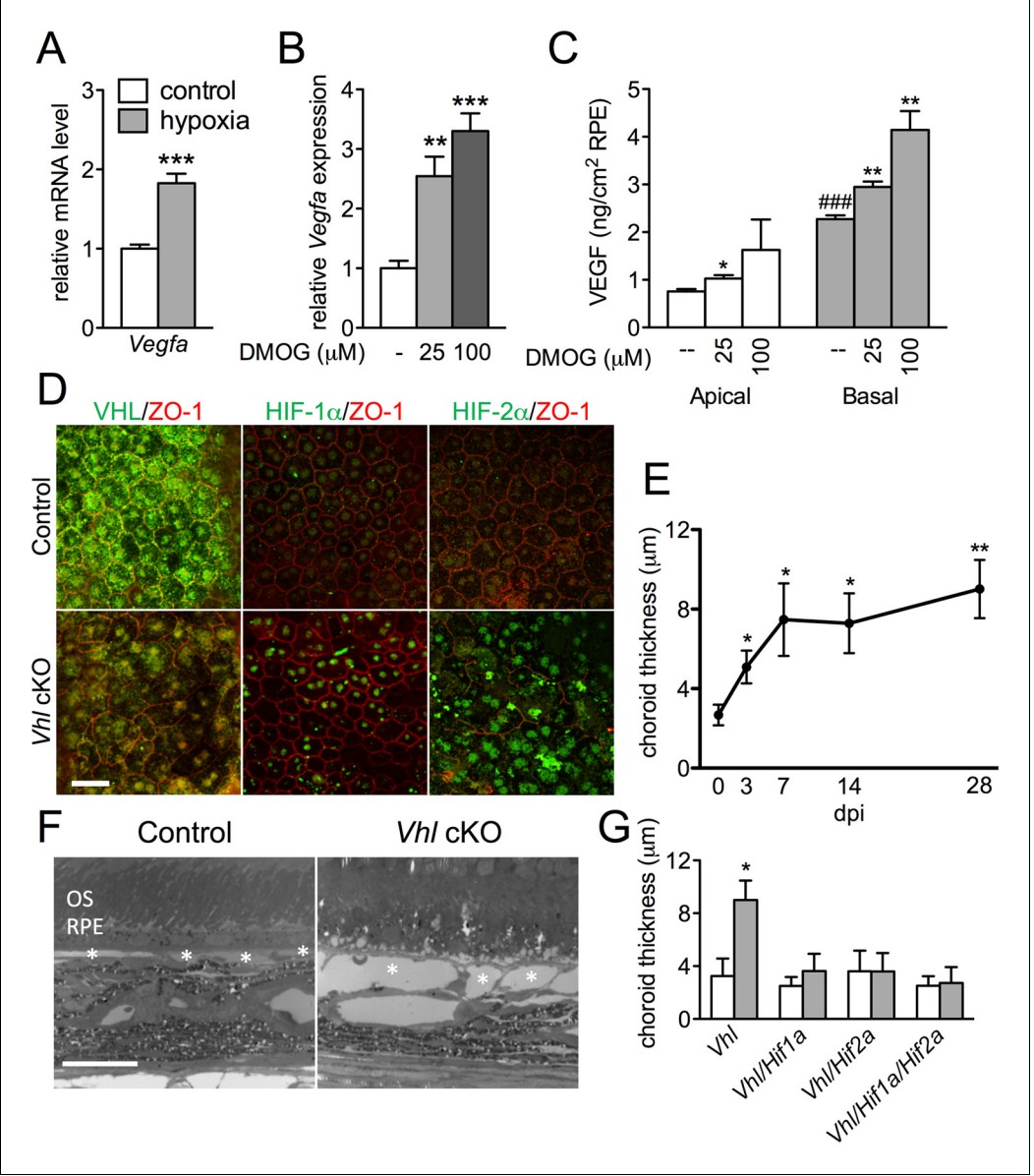

**Figure 3.** HIF-αs induce dilation of the choriocapillaris in *Vhl*-cKOs. (**A**) *Vegfa* is upregulated in hRPE exposed to 3% oxygen. Data are the mean plus s.e.m. (n=5–6). (**B–C**) *Vegfa* mRNA (**B**) and VEGF$_{165}$ protein (C—predominately from the basal surface when grown on transwells) is upregulated in a dose-dependent manner by DMOG for 24 hr compared with DMSO controls. Data are the mean plus s.e.m. (n=4–6). (**D**) Immunohistochemistry analyses reveal that VHL (green) expression is lost 3 days post induction in *Vhl*-cKOs, and HIF-1α and HIF-2α are upregulated in the nucleus of *Vhl*-cKOs RPE, but not in controls. (**E**) Measurements of the choriocapillaris from electron micrographs 0–28 days post induction (**F**) in untreated and Vhl mutants revealed progressive choriocapillaris vasodilation (n=4). (**G**) Choriocapillaris thickness values of *Vhl*-cKO, *Vhl/Hif1a*-dKO, *Vhl/Hif2a*-dKO, *Vhl/Hif1a/Hif2a*-tKO mice measured 28 days post induction (*n*=4). (See also associated *Figure 3—source data 1* for panels A-C, E, and G.) Scale bars=20 μm. Error bars represent mean plus s.d.

The following source data and figure supplement are available for figure 3:

**Source data 1.** Source data for *Figure 3A–C,E, and G*.

**Figure supplement 1.** Upregulated hypoxia-related genes in Vhl-cKO RPE/Choroid.

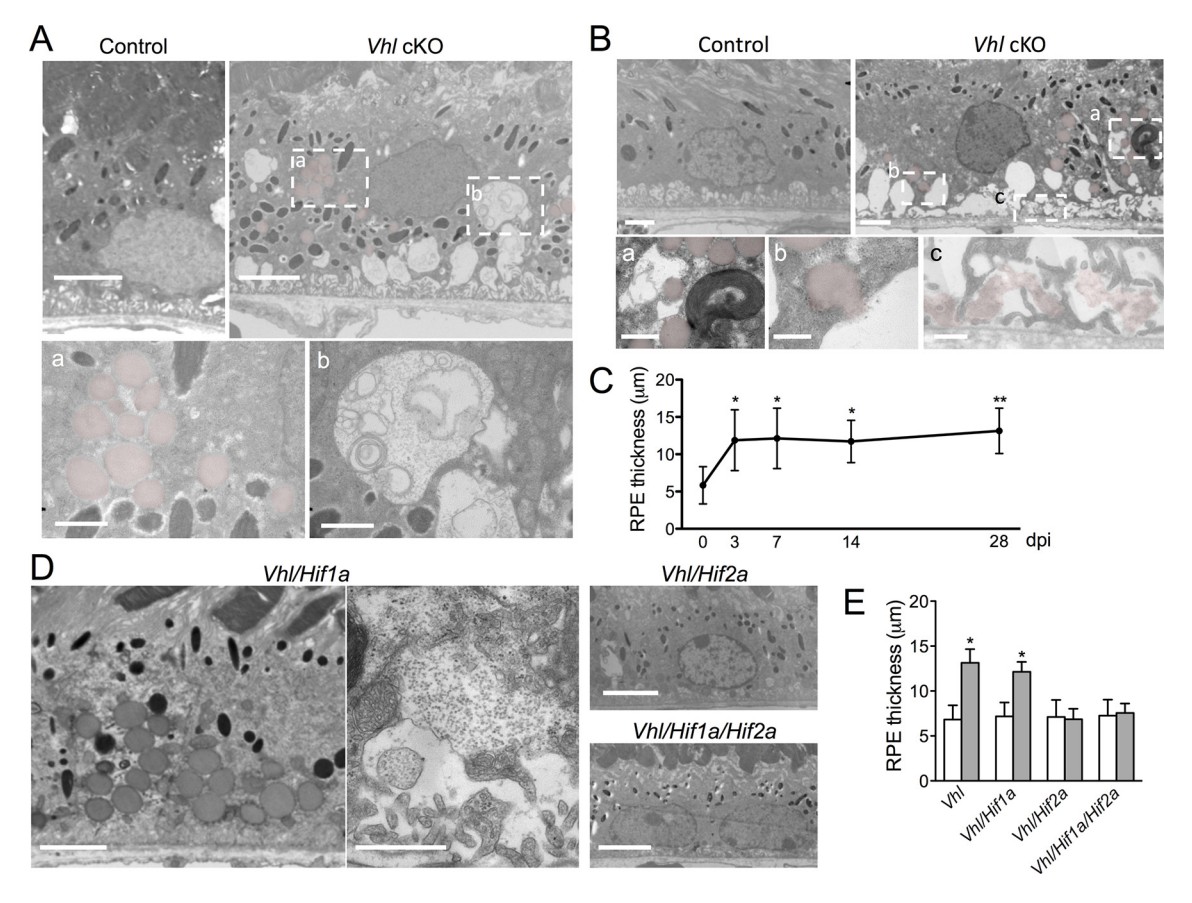

**Figure 4.** Dramatic and rapid-ensuing RPE defects observed in *Vhl*-cKO mice are dependent on *Hif2a*. (**A**) Electron micrographs of RPE cells from littermate control and *Vhl*-cKO mice 3 days post *Vhl* deletion. Regions marked with perforated white rectangles are in the lower panels. Note the intracellular accumulations of lipid droplets (a; red) and glycogen (b). (**B**) Electron micrographs of RPE cells from littermate control and *Vhl*-cKO mice 14 days post *Vhl* deletion. Intracellular lipid droplets (a), extrusion of lipid droplets into the subretinal space (b), and lipids collecting along the basal laminar surface of Bruch's membrane and between RPE basal infoldings (c) are observed. (**C**) Thickness measurements from electron micrographs and reveal that RPE hypertrophy occurs from 0–3 day post induction timepoints, and then plateaus from 3–28 timepoints in *Vhl* mutant mice (n=5). (**D**) Electron micrographs of RPE from *Vhl/Hif1a* (left panels), *Vhl/Hif2a* (upper middle panel), and *Vhl/Hif1a/Hif2a* (bottom middle panel) mutant mice 14 days post induction. Note that lipid droplets (dark gray spheres, upper left panel) and material resembling glycogen (small punctate spots) are observed in *Vhl/Hif1a*-dKO, but not in *Vhl/Hif2a*-dKO or *Vhl/Hif1a/Hif2a*-tKO RPE) 14 days post induction. These data suggest *Hif2a* is responsible for the phenotype in *Vhl* mice. (**E**) Choriocapillaris thickness values of *Vhl*-cKO, *Vhl/Hif1a*, *Vhl/Hif2a*, *Vhl/Hif1a/Hif2a* mice measured 28 days post induction (n=4). (See also associated *Figure 4—source data 1* for panels C&E.) Scale bars=5 μm (**A**), 1 μm (A'a & A'b), 2 μm (**B**), 0.5 μm (B'a, B'b, B'c), 5 μm (**D**). Error bars represent mean plus s.d.

The following source data is available for figure 4:

**Source data 1.** Source data for *Figure 4C&E*.

upon addition of dimethyloxalylglycine (DMOG), an inhibitor of prolyl hydroxylase that leads to HIF stabilization (*Figure 3B*). DMOG also significantly stimulated basal VEGF secretion in a dose dependent manner (*Figure 3C*). To determine the physiological relevance of hypoxia-enhanced RPE-derived VEGF synthesis, we used *VMD2-Cre* to delete *Vhl* in RPE in vivo. Signs of [pseudo] hypoxia, i.e. HIF-α immunoreactivity and activation of known hypoxia-induced genes (including *Vegfa*), were observed three days post induction in *Vhl*-cKO mice (*Figure 3D* and *Figure 3—figure supplement 1*). The increased VEGF production from *Vhl*-cKO RPE correlates with significant and progressive choriocapillaris vasodilation based on ultrastructural examinations (*Figure 3E and F*). However, no vasodilation was observed in double (*Vhl/Hif1a*-dKO or *Vhl/Hif2a*-dKO) or triple (*Vhl/Hif1a/Hif2a*-tKO) knockout mice (*Figure 3G*) indicating that both RPE-derived HIF-1α and HIF-2α are mutually

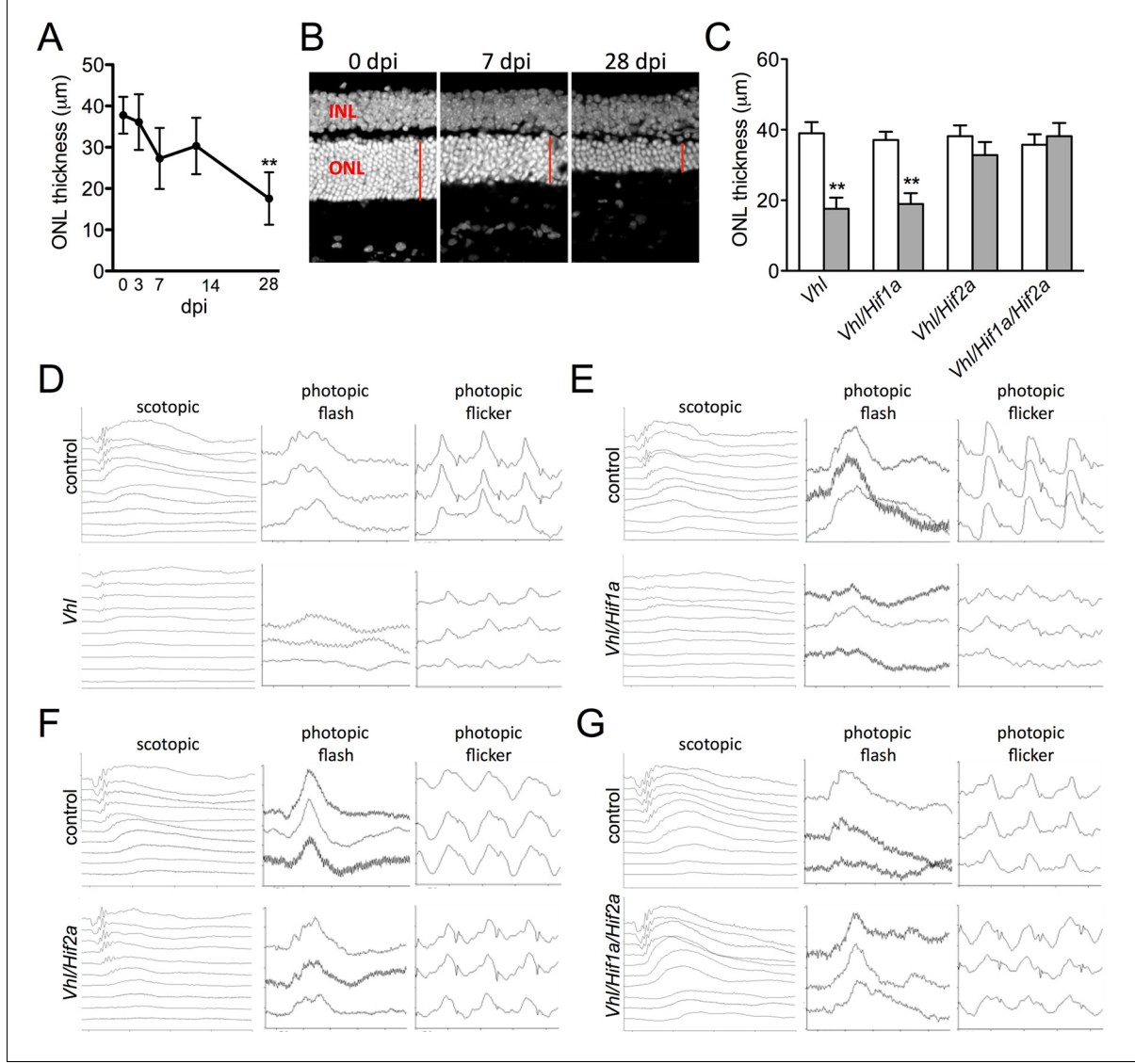

**Figure 5.** Progressive and rapid photoreceptor degeneration observed in *Vhl*-cKO mice is dependent on *Hif2a.* (**A**) Thickness measurements from electron micrographs reveal progressive thinning of the outer nuclear layer (ONL or photoreceptor cell bodies) from 0–28 days post induction timepoints in *Vhl* mutant mice (n=5). (**B**) Cryosectioned DAPI stained retinas from *Vhl*-cKO mice prior to induction (0 dpi; left panel), 7 dpi, and 28 dpi. (**C**) Quantified thickness values measured 600 μm from the optic nerve head of the outer nuclear layer in *Vhl*-cKO, *Vhl/Hif1a*, *Vhl/Hif2a*, and *Vhl/Hif1a/ Hif2a* 28 days post induction (*n*=4) (See associated *Figure 5—source data 1* for panels A&C.). (**D**) Full-field ERGs performed on *Vhl*-cKO and control mice 28 days post induction. (**E**) ERGs from *Vhl/Hif1a*, (**F**) *Vhl/Hif2a*, and (**G**) *Vhl/Hif1a/Hif2a* mutant mice 28 days post induction. ERG analyses reveal that normal photoreceptor function is observed in *Vhl/Hif2a* (**F**) or *Vhl/Hif1a/Hif2a* (**G**) mutant mice. *p<0.05, **p<0.01. For all ERGs n=6–8. Error bars indicate mean plus s.d.

The following source data and figure supplements are available for figure 5:

**Source data 1.** Source data for *Figure 5A&C* and *Figure 5—figure supplement 1A–C*.

**Figure supplement 1.** Quantification of ERGs for *Vhl* cKO and lack of photoreceptor degeneration in relevant controls.

**Figure supplement 2.** RPE and choroid defects characteristic of retinal remodeling are observed in late stage *Vhl* mutants.

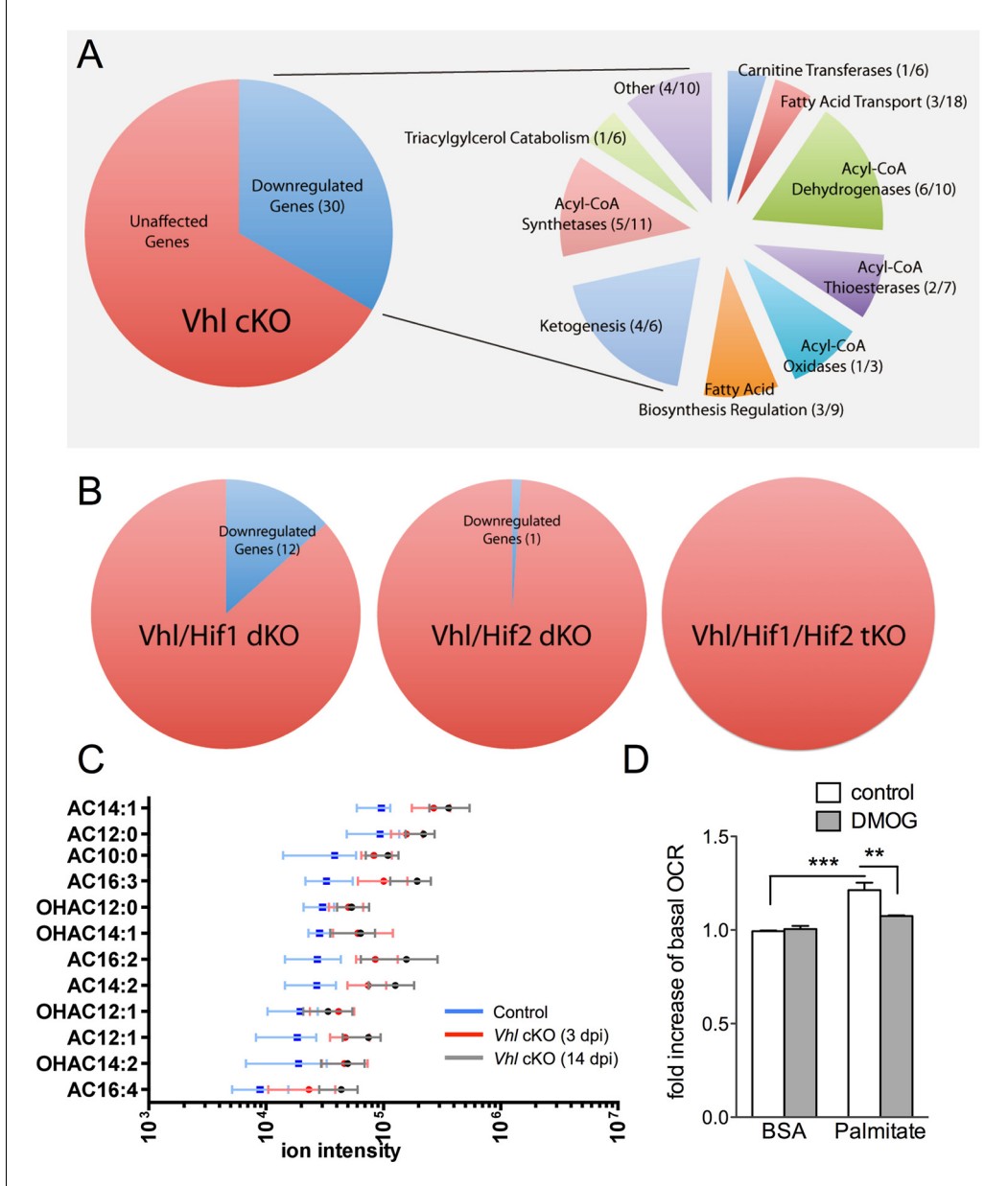

**Figure 6.** Defects in lipid metabolism in *Vhl* mutant RPE. (**A**) Summary of gene-profiling experiments for lipid metabolism genes in RPE/choroid complexes from *Vhl* mutant mice 3 days post induction (n=4). (**B**) Downregulation of lipid metabolism genes was also seen in *Vhl/Hif1a* mutants, but nominally in *Vhl/Hif2a*, and no gross changes were seen in *Vhl/Hif1a/Hif2a* mutants 3 days post induction (n=4). (**C**) Untargeted high-resolution mass spectrometry-based metabolomic analyses revealed that a group of acylcarnitines (AC) was progressively dysregulated from 3 to 14 days post induction (*n*=4–6) (see also associated *Figure 6—source data 1*). Box and whiskers plots are shown. Error bars represent maximum and minimum values. (**D**) Pre-treating hRPE with DMOG reduced the basal oxygen consumption rates (initial OCR – OCR after injection of RAA) when the cells were assayed in substrate limited media (2.5 mM glucose) and provided BSA control or palmitate conjugated to BSA (n=4) (see also associated *Figure 6—figure supplement 3B*). Error bars are the maximum and minimum values in panel C and mean plus s.d. in panel D.

The following source data and figure supplements are available for figure 6:

**Source data 1.** Source data for *Figure 6C*, *Figure 6—figure supplement 2A–D*, *Figure 6—figure supplement 3B*.

*Figure 6 continued on next page*

*Figure 6 continued*

**Figure supplement 1.** Classification of lipid metabolism genes and gene-profiling in *Vhl* and combinatorial *Vhl/Hif* mutants.

**Figure supplement 2.** MS-based metabolomic assays for *Vhl*-cKO, *Vhl/Hif1a*-dKO, *Vhl/Hif2a*-dKO, *Vhl/Hif1a/Hif2a*-tKO mice.

**Figure supplement 3.** DMOG inhibits lipid oxidation in RPE cells.

responsible for HIF/VEGF induced vasodilation. Collectively, these findings demonstrate that RPE can sense changes in oxygen/nutrient availability and respond by appropriately altering the vascular tone of the choriocapillaris.

## Hypoxia induces structural changes in RPE and retinal degeneration

Hypoxia-triggered choriocapillaris vasodilation may come at a significant cost for the retina. The accumulation of lipid droplets and material resembling glycogen is detectable within only three days post induction in the RPE of *Vhl* mutants (*Figure 4A*). Severely distended basal infoldings, thickening of Bruch's membrane, and numerous lipid droplets (sometimes contiguous with subretinal extracellular spaces) are observed 14 days post *Vhl* deletion (*Figure 4B*; red). Measurements across the RPE from electron micrographs reveal significant hypertrophy (*Figure 4C*). While the double deletion of *Vhl/Hif1a* did not prevent the RPE defects in the *Vhl* mutants, the RPE cells of *Vhl/Hif2a* or *Vhl/Hif1a/Hif2a* mutants appeared unremarkable and were not hypertrophic (*Figure 4D and E*), suggesting that HIF-2α, as it is in other cell-types (*Qiu et al., 2015*; *Zhao et al., 2015*), is the pathological HIF isoform in hypoxic RPE.

Devastating secondary effects resulting from RPE hypoxia were observed in the sensory retina. Photoreceptor degeneration (*Figure 5A–C*) and significant functional impairments of both rod and cone driven-pathways (*Figure 5D—figure supplement 1A*) were observed in *VMD-Cre;Vhl* and *VMD-Cre;Vhl/Hif1a* mice 28 days post induction but not in in *Vhl/Hif2a* (or *Vhl/Hif1a/Hif2a* mice; *Figure 5C,E–G*), (or in any of the relevant controls; *Figure 5—figure supplement 1B and C*). In advanced stages of the phenotype (>50 dpi), dramatic changes in RPE and the vasculature are observed consistent with retinal remodeling (*Figure 5—figure supplement 2*) (*Marc et al., 2003*). These findings imply that HIF-2α-mediated metabolic stress in RPE, which cannot be rescued even with a significantly dilated choriocapillaris, is enough to promote photoreceptor degeneration.

## Lipid handling is impaired in hypoxic RPE

We next set out to identify the molecular mechanisms driving the hypoxic-mediated metabolic stress in RPE cells. Based on histopathological evidence of impaired lipid handling in hypoxic RPE we performed gene profiling for fatty acid metabolism genes from RPE/choroid complexes of *Vhl*-cKO mice. Acyl-CoA synthetase and Acyl-CoA dehydrogenase family genes were downregulated (*Figure 6A* and *Figure 6—figure supplement 1A and B*) but not in *Vhl/Hif2a* or *Vhl/Hif1a/Hif2a* mutant RPE (*Figure 6B* and *Figure 6—figure supplement 1B*). We also performed untargeted high-resolution mass spectrometry-based metabolomic analyses and observed abnormal levels of several long-chain saturated, unsaturated, and oxidized acylcarnitines in the *Vhl*-cKO mice (*Figure 6C* and *Figure 6—figure supplement 2*). *Vhl/Hif1a* mice exhibit dysregulated metabolomic profiles similar to *Vhl* mutants (*Figure 6—figure supplement 2A and B*), but normal levels of acylcarnitines and other metabolites were observed in *Vhl/Hif2a* (*Figure 6—figure supplement 2C*) and *Vhl/Hif1a/Hif2a* mice (*Figure 6—figure supplement 2D*). Collectively, these data strongly suggest that HIF-2α regulates lipid handling in RPE in vivo.

To test if HIF activation leads to altered lipid oxidation we monitored oxygen consumption in human RPE treated with a hypoxia mimetic, DMOG. Seahorse flux analysis revealed that human RPE cells in substrate-limited media oxidize exogenous lipids (palmitate) as an energy substrate (BSA control vs. palmitate control, *Figure 6D*). The ability of palmitate to increase oxygen consumption rates (OCRs) was validated by the addition of an inhibitor of the carnitine transport, etomoxir, which reduced oxygen consumption to BSA control levels (*Figure 6—figure supplement 3A*).

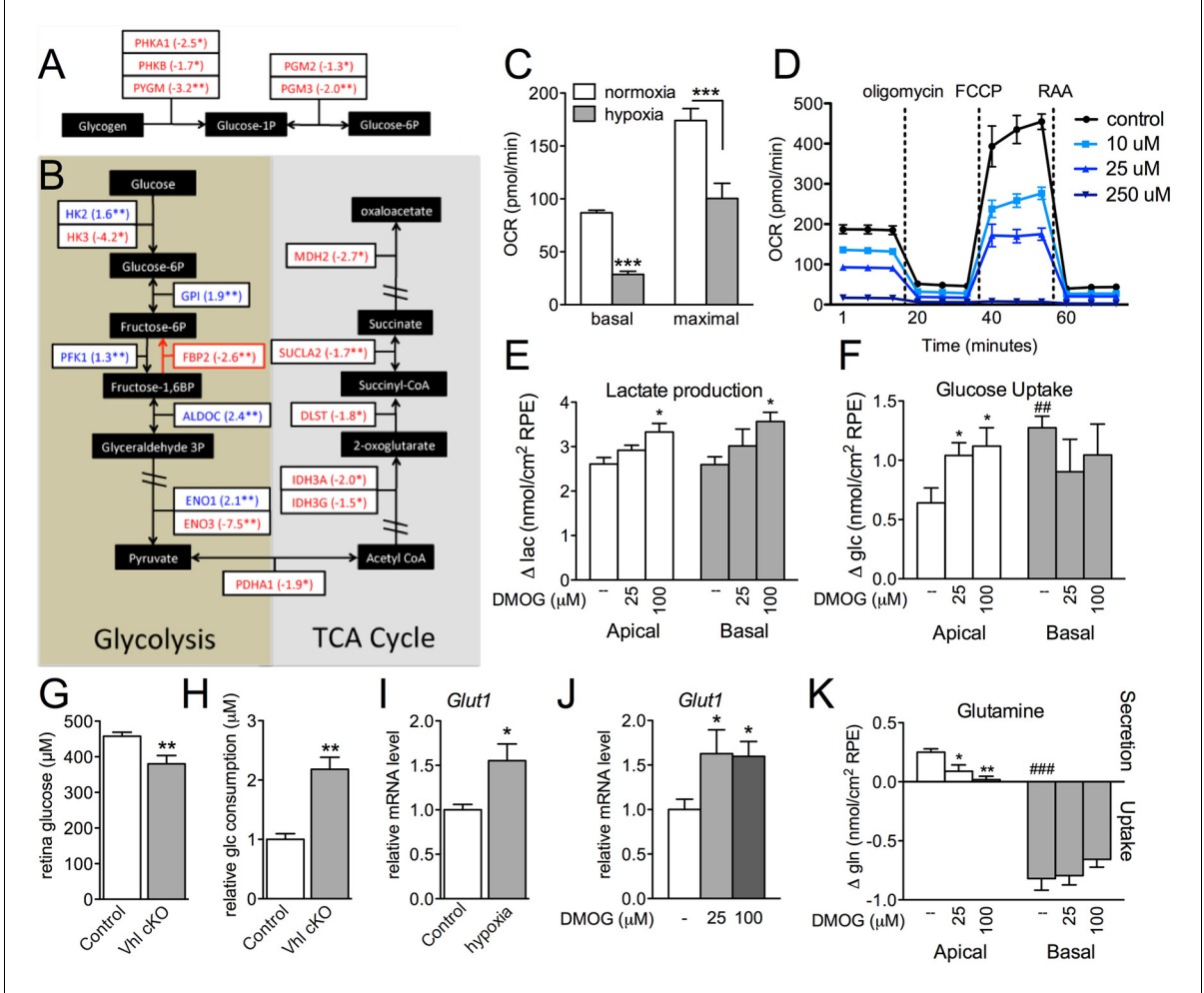

**Figure 7.** Glucose consumption and metabolism is altered in the RPE of *Vhl* mutants. (**A**) Gene-profiling data revealed that glycogen degradation genes were significantly attenuated in *Vhl* mutants 3 days post induction (n=4). (**B**) Gene-profiling data for glucose metabolism genes were summarized by plotting changes along the glycolysis and TCA cycle pathways 3 days post induction (n=4) (red text=downregulated, blue text=upregulated). (**C**) Basal and maximal oxygen consumption rates (OCR) of hRPE after being exposed to 3% $O_2$ for 72 hr or maintained at normoxia. Data are the mean plus s.e.m. (n=7–10). (**D**) Seahorse Flux Analysis OCR trace showing reduced OCR in hRPE cells treated with DMOG for 24 hr. Data points are the mean plus s.d. (n=6). (**E and F**) Changes in lactate (**E**) and glucose (**F**) levels of the media from hRPE cells, in transwells, after treatment with DMOG for 24 hr. Data are the mean plus s.e.m. (n=4). (**G**) Glucose levels are decreased in the sensory retina of *Vhl*-cKO mice 3 days post induction compared with littermate controls (*n*=6–10). (**H**) Relative glucose consumption is increased (roughly two-fold) in primary *Vhl*-cKO RPE compared with controls (*n*=6). (**I and J**) Relative expression level of *Glut1* in hRPE cells after exposure to 3% $0_2$ for 72 hr (**I**) or treatment with DMOG for 24 hr (**J**). Data are the mean plus s.e.m. (n=6). (**K**) Changes in glutamine levels of the media from hRPE cells, in transwells, after treatment with DMOG for 24 hr. (See also associated *Figure 7—source data 1* for panels C-K.) Data are the mean plus s.d. (n=4).

The following source data and figure supplements are available for figure 7:

**Source data 1.** Source data for *Figure 7C–K*, *Figure 7—figure supplement 2A*.

**Figure supplement 1.** Altered regulation of glycolytic and TCA genes in *Vhl* and *Hif* mutants.

**Figure supplement 2.** Glucose metabolism is altered in *Vhl* mutant RPE and hRPE exposed to hypoxia.

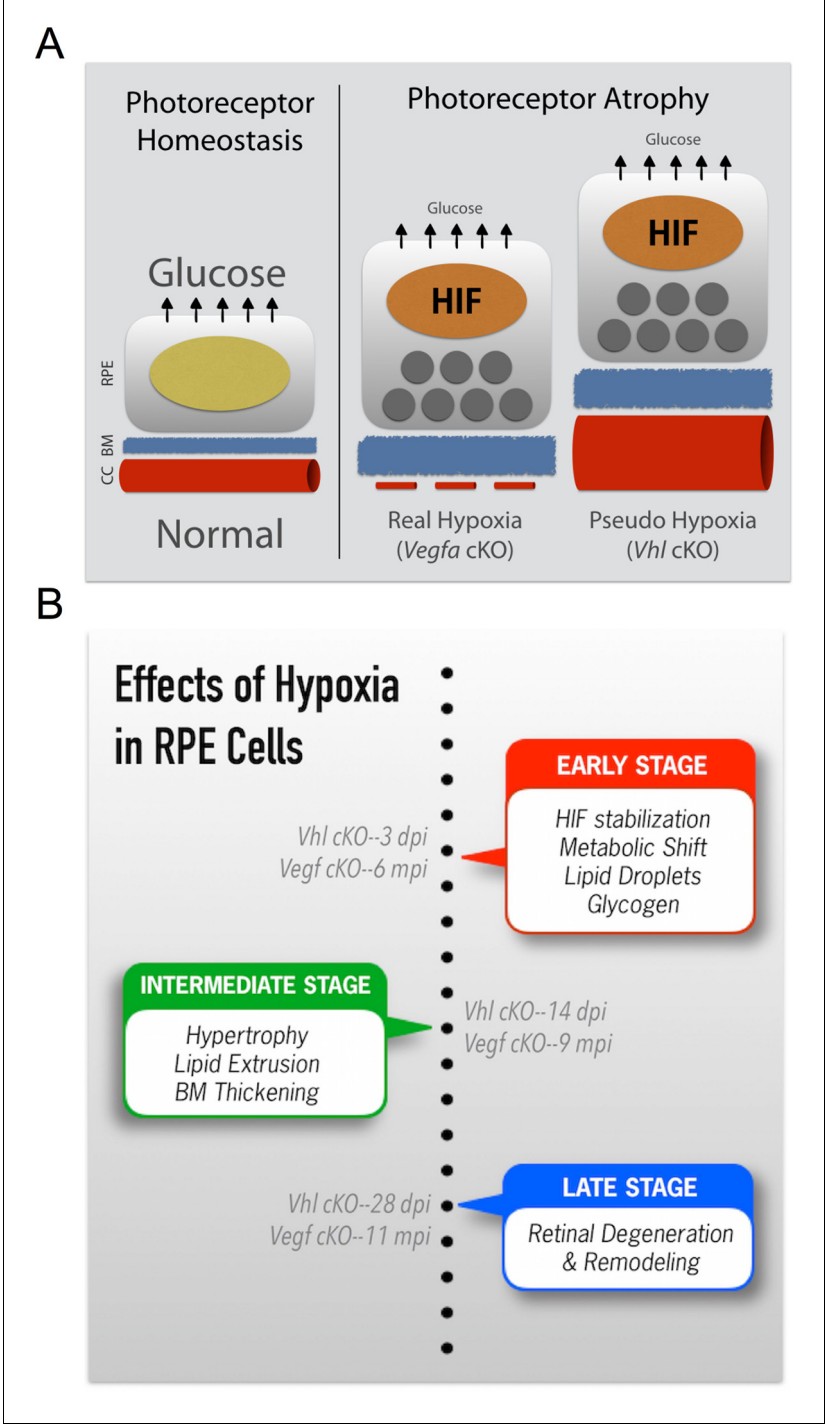

**Figure 8.** Summarizing the onset of the major phenotypes in *Vhl* and *Vegfa* mutant mice. (**A**) In normal conditions RPE deliver sufficient levels of glucose for photoreceptor homeostasis. During real (*Vegfa*-cKO) and pseudo hypoxia (*Vhl*-cKO), Bruch's membrane thickening and metabolic changes in RPE (that induce lipid accumulation—dark gray circles) limit glucose delivery to photoreceptors. This results in photoreceptor atrophy. (**B**) The effects of pseudo hypoxia in *Vhl* mutants and real hypoxia in *Vegfa* mutants present in a similar sequence but at radically different rates. The progression rate is accelerated in *Vhl*-cKOs since HIFs are dominant-stable. Changes in choriocapillaris density induce graded hypoxia in the *Vegfa*-cKO line, making it more physiologically relevant.

Interestingly, treating the cells with low-dose DMOG for 48 hr prevented lipid oxidation (palmitate control vs. palmitate DMOG, *Figure 6D* and *Figure 6—figure supplement 3B*). These data suggest that hypoxic RPE alter their lipid handling behaviors and begin storing lipids in droplets, rather than utilizing them as an energy substrate.

## Glucose metabolism is disrupted in hypoxic RPE

The presence of visible glycogen stores in *Vhl*-cKO RPE indicates that glucose metabolism may also be dysregulated. Using PCR arrays for glucose metabolism we determined that glycogen degradation genes were downregulated in *Vhl* mutant mice (summarized in *Figure 7A*). Furthermore, we observed that glycolysis-related genes were largely upregulated and TCA cycle genes were largely downregulated (summarized in *Figure 7B* and *Figure 7—figure supplement 1A and B*). These data suggest that in vivo the RPE cells are reducing oxidative phosphorylation and meeting their energy demands by increasing glycolysis. To test if RPE cells reduce oxidative metabolism in response to hypoxia, we monitored oxygen consumption rates in cultured human RPE cells (provided glucose, pyruvate and glutamine) after the cells were exposed to hypoxic conditions (3% $O_2$) for 72 hr compared to a parallel control plate maintained in normoxia. Basal and maximal oxygen consumption rates were greatly reduced in cells after hypoxia exposure (*Figure 7C* and *Figure 7—figure supplement 2A*), suggesting mitochondrial respiration has been remodeled as a result of hypoxia. An internal ratio of the basal OCR divided by the proton leak (oligomycin-insensitive OCR) to normalize for potential plate-to-plate variation, showed a similar effect of hypoxia reducing oxidative capacity (*Figure 7—figure supplement 2B*). Treating the human RPE cells with low-dose DMOG, a HIF activator, also significantly reduced basal and maximal oxidative capacities in a dose-dependent manner (*Figure 7D*). In cells treated with higher doses of DMOG ($\geq$250 uM) oxidative capacity was completely lost (*Figure 7D*), even though there were no outward signs of toxicity or cell death. Collectively these data suggest that the oxidative metabolism of RPE cells is very sensitive to hypoxia.

We found using 3D culture methods that polarized human RPE treated with DMOG increased lactate production (*Figure 7E*) further supporting the notion of a metabolic shift from oxidative phosphorylation toward glycolysis in RPE. This was coupled with evidence of increased glucose uptake from their apical surfaces (*Figure 7F*), thereby potentially depleting glucose supplies allocated for the photoreceptors. In vivo, metabolic profiling analyses revealed that glucose levels in the sensory retina of *Vhl* mutants were significantly reduced (*Figure 7G*). Similarly, primary RPE cells isolated from Vhl mutant mice took up more than double the amount glucose as controls (*Figure 7H*). To allow increased glucose uptake, we found the glucose transporter 1 (Glut1 or Slc2a1) was upregulated in vivo (*Vhl*-cKO RPE; *Figure 3—figure supplement 1*) and in vitro in response to both hypoxia and DMOG treatment (human RPE *Figure 7I and J*). These changes in glucose processing are similar to a HIF-dependent phenomenon observed in tumor cells known as the 'Warburg effect' (*Luo et al., 2011*; *Takubo et al., 2010*; *Warburg, 1956*).

Measuring metabolite levels from human RPE differentiated in transwells highlights how changes in RPE metabolism may influence nutrient availability for the retina. In addition to DMOG reducing glucose (and increasing lactate) levels in the apical chamber, we observed that glutamine handling was also altered in response to hypoxia. Glutamine is an essential cellular metabolite that is both a key energy substrate and nitrogen source. DMOG reduced glutamine availability in the apical chamber (*Figure 7K*). These data suggest that a confluence of hypoxia-mediated metabolic changes in RPE results in depleted energy substrates for photoreceptors that promotes retinal degeneration.

## Discussion

Photoreceptors are some of the most metabolically demanding cells in the body; this study reinforces how heavily photoreceptors rely on RPE for metabolic support. While it is widely accepted that death or dysfunction or RPE results in photoreceptor degeneration, the combination(s) of cellular stresses that impair RPE function and survival are incompletely understood. We present evidence here that a single causative factor, chronic HIF-2α-mediated metabolic stress in RPE cells, is enough to induce photoreceptor dysfunction/degeneration. We also provided the following mechanistic explanations: (a) hypoxia alters the RPE secretome and vectorial release of VEGF, lipids, and several metabolites including lactate, glutamine, and glucose; (b) hypoxic RPE shut down glucose and lipid oxidation and commit to glycolysis (and store superfluous glucose and lipid); (c) this metabolic shift

allows RPE to obtain energy more quickly, but requires that they double their glucose intake due to low ATP yield. Since RPE are the major suppliers of glucose to the neurosensory retina (*Foulds, 1990*), these changes directly impact photoreceptor function and survival (*Linton et al., 2010*; *Chertov et al., 2011*).

Aberrant lipid accumulation occurs in both human RPE and Bruch's Membrane during aging and disease (*Pauleikhoff et al., 1990*; *Holz et al., 1994*; *Sheraidah et al., 1993*). In fact lipids were the first identified molecules in human subretinal deposits (*Pauleikhoff et al., 1992*; *Wolter and Falls, 1962*). Unlike retinosomes, lipid droplets that arise due to visual cycling defects in RPE (*Imanishi et al., 2004*), the lipid inclusions observed in hypoxic RPE are likely derived from metabolic derangements. Similar inclusions (and RPE hypertrophy and photoreceptor atrophy) were observed in transgenic mice by conditionally inactivating OXPHOS in RPE (*Best1-Cre;Tfam$^{fl/fl}$*) (*Zhao et al., 2011*). Lipid accumulation in Bruch's membrane is a common feature of AMD, and several lipid-processing genes have been associated with modifiable risk for AMD (*Chen et al., 2010*; *Klaver et al., 1998*; *Neale et al., 2010*; *Souied et al., 1998*; *Zerbib et al., 2009*). Neutral lipids, likely derived from RPE, are abundant in AMD specific basal linear deposits (*Sarks et al., 1988*; *Curcio and Millican, 1999*; *Rudolf et al., 2008*; *Sarks et al., 2007*; *Zweifel et al., 2010*). Finally, thickening of Bruch's membrane contributes to AMD pathogenesis (*Sarks, 1976*; *Sarks et al., 1988*), probably by greatly limiting diffusion of aqueous based metabolites to the retina (including energy substrates such as glucose) (*Pauleikhoff et al., 1990*; *Moore, 1995*; *Starita et al., 1995*; *1996*; *1997*), by inducing inflammation, and by promoting choroidal neovascularization (*Baba et al., 2010*; *Tamai et al., 2002*). The identification of a broad class of acyl-carnitines in hypoxic RPE provides additional evidence of gross glucose metabolism and lipid-handling defects; identifying the combinations of perturbed acylcarnitines is informative for diagnosing fatty acid oxidation defects and other inborn errors of metabolism (*Rinaldo et al., 2008*).

Glucose availability correlates with retinal degeneration in both animal models and AMD patients. A comprehensive analysis of murine models of spontaneous degeneration revealed nutritional imbalances leading to cone photoreceptor degeneration, which could be delayed with insulin therapy (*Punzo et al., 2009*). In addition RPE-specific inactivation of oxidative phosphorylation, which is induced by conditional deletion of mitochondrial transcription factor A, leads to increased glycolysis and photoreceptor degeneration. These phenotypes were prevented by inhibition of mTOR (mammalian target of rapamycin) (*Zhao et al., 2011*), a positive regulator of HIF-αs (*Hudson et al., 2002*; *Nayak et al., 2012*). Furthermore, excessive glycemic load is a modifiable risk for AMD; and 20% of advanced AMD cases might be prevented by consuming less glucose-rich foods (*Chiu et al., 2007*).

Hypoxia in RPE can occur during aging and disease due to localized choriocapillaris dropout. Based on this study, morphometric changes to the choriocapillaris, which can be routinely and non-invasively examined (*Zhang et al., 2015*), may be a powerful predictive tool for AMD disease progression. In addition, monitoring RPE/Bruch's membrane changes and tracking the glycemic index of early AMD patients may improve case-specific treatment and risk progression prediction algorithms. Finally, abnormally high HIF-2α levels have been observed in the RPE of human aged donor eyes (*Sheridan et al., 2009*). Therefore, anti-HIF therapies, which are being developed for treating other diseases (*Metelo et al., 2015*; *Rini and Atkins, 2009*; *Rogers et al., 2013*; *Scheuermann et al., 2013*), may be effective for treating some AMD patients. In conclusion, our models may be employed to examine the progression of AMD-like features, as well as for developing novel preventive and therapeutic strategies for AMD and other vision-threatening diseases.

## Materials and methods

### Mice

The TSRI Animal Care Committee approved all procedures involving animals and we adhered to all federal animal experimentation guidelines. Transgenic mice carrying the human vitelliform macular dystrophy-2 (VMD2) promoter-directed reverse tetracycline-dependent transactivator (rtTA) and the tetracycline-responsive element (TRE)-directed Cre recombinase (*VMD2-Cre* mice) (*Le et al., 2008*) were mated with *Vhl$^{fl/fl}$* mice (*Haase et al., 2001*), *Hif1a$^{fl/fl}$* mice (*Ryan et al., 2000*), *Epas (Hif2a)$^{fl/fl}$* mice (*Gruber et al., 2007*), or *Vegfa$^{fl/fl}$* mice (*Gerber et al., 1999*). Control littermates harboring floxed alleles but no Cre-recombinase were utilized. Additional control experiments were also

examined (*Figure 5—figure supplement 1B and C*). To induce gene deletion, 80 µg/g body weight of doxycycline were injected daily into 6–8 week-old transgenic mice intraperitoneally for three days. Gene recombination was quantified and cre-mediated toxicity was examined in double transgenic *VMD2-Cre* with *ROSA26 mTomato/mGFP* reporter mice (*Muzumdar et al., 2007*). We noted that recombination in *VMD2-Cre* occurs at either ~70% or ~10%, and either the high or low efficiency is inherited. All data shown in the main body of text was from the offspring of "high efficiency mice." Gene ablation in "low efficiency mice" yielded a similar, albeit much milder phenotype. Genotyping was performed at Transnetyx (Memphis, TN). Mutations for *rds (retinal degeneration slow), rd1, rd8*, and *rd10* were examined and excluded from breeding pairs.

## Primary RPE cell culture

RPE cells from *VMD2-Cre;Vhl^{fl/fl}* and control *Vhl^{fl/fl}* littermates were isolated as previously described (*Krohne et al., 2012*). Cells were maintained at 37° and 5% $CO_2$ in DMEM/F12 from Thermo Fisher Scientific (Waltham, MA) with 2% FBS from Jackson ImmunoResearch (West Grove, PA). 2 µg/µl doxycycline was added to all cultures daily for three days to induce *Vhl* ablation. Human RPE cells (Lonza) were maintained either on plastic surfaces or on transwell filters (Corning) depending on the application in RtEGM Retinal Pigment Epithelial Cell Growth Medium fortified with RtEGM BulletKit from Lonza (Basel, Switzerland). Glucose, lactate, and glutamine levels were measured from the media in apical and basal compartments using a 2900 Biochemistry Analyzer from YSI (Yellow Springs, OH). A concentrated stock of DMOG from Cayman Chemical (Ann Arbor, MI) was made with DMSO, and added directly to the media in different concentrations to induce pseudo-hypoxia. RPE cells were maintained in low oxygen and metabolic changes were analyzed using Seahorse Flux Analysis (see below) in a Coy Dual Hypoxia Chamber from Coy Lab Products (Grass Lake, MI) (*Grassian et al., 2014*).

## Glucose assays

In vitro glucose consumption analyses (GAHK20-1KT, Sigma-Aldrich; St. Louis, MO) and in vivo glucose measurements (SKU120003, Eton Bioscience Inc.; San Diego, CA) were performed on the fourth day after doxycycline administration according to the manufacturer's instructions. For in vivo glucose measurements, retinas were dissected out and homogenized in 80% ethanol.

## Seahorse flux analysis

RPE were cultured on 96 well XF Microplates (Agilent Technologies Inc.; La Jolla, CA)) and maintained as described above. Prior to analysis human RPE were transferred to assay media (modified Ringer's solution lacking sodium bicarbonate). For glucose metabolism assays media was supplemented with 12 mM glucose, 2 mM pyruvate, 2 mM glutamine, and 10 mM HEPES, pH 7.4. For fatty acid oxidation (FAO) assays the media contained 2.5 mM glucose, 0.5 mM carnitine and 10 mM HEPES, pH 7.4. Oxygen consumption rates (OCR) were measured using a XF^e96 device (Agilent Technologies Inc.)); the following concentrations of drugs were used: 2 µM oligomycin, 0.5 µM carbonyl cyanide p-trifluoromethoxyphenylhydrazone (FCCP), 1 µM antimycin A, and 1 µM rotenone. For measuring fatty acid oxidation higher concentrations of drugs were used: 2 µM oligomycin, 1 µM FCCP, 4 µM antimycin A, and 2 µM rotenone. For FAO, etomoxir (40 µM/well) was added at least 30 min prior to analysis. BSA and BSA-conjugated palmitate (Agilent Technologies Inc.) were added to the cells immediately prior to reading. Basal OCR was calculated by subtracting the OCR in the presence of antimycin A and rotenone from that of the initial OCR prior to addition of drugs. Maximal OCR was calculated by subtracting OCR in the presence of antimycin A and rotenone from the OCR in the presence of FCCP.

## Electron microscopy analysis

Cardiac perfusions were performed with freshly prepared saline, and then with 4% paraformaldehyde plus 1.5% gluteraldehyde in 0.1 M cacodylate buffer. Eyes were enucleated and fixed in the same mixture overnight at 4°C and then rinsing in 0.1 M Na cacodylate buffer for 1 hr. The eye cups were then postfixed in 1% OsO4 in 0.1 M cacodylate buffer for 2 hr, followed by another 1-hr wash and dehydration with graded ethanol solutions. Samples were incubated overnight in a 1:2 mixture of propylene oxide and Epon/Araldite (Sigma-Aldrich) and then placed in 100% resin followed by

embedding. The blocks were sectioned and used for high-magnification electron microscopy analysis.

## Electroretinography

Fully dark adapted mice were anesthetized under dim red light by intraperitoneal injection of 15 mg/kg ketamine and 7 mg/kg xylazine. Full-field ERGs were recorded from the corneal surface of each eye after pupil dilation (1% tropicamide and 2.5% phenylephrine) using active contact lens electrodes (Mayo; Inazawa, Japan) placed on the cornea, a mouth reference, and tail ground electrode. A computerized system with an electronically controlled Ganzfeld dome was used (Espion E2 with Colordome; Diagnosys; Westford, MA). In the dark-adapted state, we recorded rod and mixed cone/rod responses to a series of white flashes of increasing intensities ($1 \times 10^{-5}$ to 50 cd•s/m$^2$). In the light-adapted state, with a 30 cd/m$^2$ background, cone responses to 1-Hz (0.63 to 20 cd•s/m$^2$) and 30-Hz (3.98, 10, and 20 cd•s/m$^2$) flicker stimuli were recorded. All ERG responses were filtered at 0.3–500 Hz, and signal averaging was applied.

## Immunohistochemistry

Retinas or RPE/choroid complexes were dissected and prepared for whole mounts or sectioning. For preparation of retinal cross-sections, dissected retinas were laid flat with 4 radial relaxing incisions, placed in 4% PFA, and incubated at 4°C overnight. Retinas were then placed in 20% sucrose at 4°C for 4 hr and embedded in Tissue-Tek OCT compound (Sakura Finetek; Torrance, CA) for cryosectioning. The entire retinas were sectioned, and central sections from at least three retinas from at least three mice were examined. Primary antibodies were used including anti- ZO-1 (Life Sciences, Carlsbad, CA), HIF-1α (Novus Biologicals; Littleton, CO), HIF-2α (Novus Biologicals), and VHL (Santa Cruz Biotechnology; Santa Cruz, CA). Fluorescent-conjugated isolectin Griffonia Simplicifolia IB-4 (Lectin) was also used (Life Sciences). Images were obtained using a confocal microscope (LSM710, Carl Zeiss; Oberkochen, Germany)

## In vivo imaging

Color fundus images were captured using a Micron III platform (Phoenix Research Laboratories; Pleasanton, CA). Confocal scanning laser ophthalmoscopy (cSLO) detecting fundus autofluorescence, indocyanine green angiography, and optical coherence tomography (OCT) were performed using Spectralis (Heidelberg Engineering; Heidelberg, Germany) and Envisu (Bioptigen; Durham, NC) instruments.

## Quantification of retinal thickness

Retinal thickness values of the ONL, OS, RPE, and choriocapillaris were measured at 600 μm from the optic nerve head from electron micrographs using NIH ImageJ software. RPE thickness measurements were made by measuring the area between end of the basal infoldings and the outer segments of the photoreceptors.

## Proteomic and transcriptomic arrays

Proteomic arrays for angiogenesis-related proteins (Angiogenesis Proteome Profiling Array, R&D Systems, Inc; Minneapolis, MN) and ELISAs for human VEGF (R&D Systems) were performed according to the manufacturer's instructions. For mRNA arrays, total RNA was prepared from RPE/choroid complexes using the RNeasy Plus Mini kit (Qiagen; Hilden, Germany) and was reverse transcribed using the RT$^2$ First Strand cDNA Kit (Qiagen). mRNA PCR arrays for hypoxia signaling, glucose metabolism, and fatty acid metabolism (RT$^2$ Profiler PCR Array for Mouse Hypoxia Signaling Pathway (PAMM-032), Mouse Glucose Metabolism (PAMM-006), and Mouse Fatty Acid Metabolism (PAMM-007), Qiagen) were performed according to the manufacturer's instructions. Quantitative PCR assays were performed on a real-time PCR System (ABI 7900HT Fast; Thermo Fisher Scientific).

## LC-MS/MS based untargeted metabolomics

After harvesting, eyes were flash frozen in LN$_2$ and stored at -80°C prior to processing. Eyes were lyophilized overnight. The next day, 500 μL of 10% CHCl$_3$ in MeOH was added to each tube. The eyes were subjected to 5 rounds of vortexing for 30s, frozen in LN$_2$ and sonicated for 10 min at

50°C. The samples where then incubated for 1 hr at −20°C and centrifuged for 10 min @ 13,000 rpm, 4°C in a microcentrifuge. The supernatant was collected in a clean vial, an additoinal 500 μL of 10% $CHCl_3$ in MeOH was added to each eye, and the extraction procedure was repeated. The supernatants were pooled and dried in a centrifugal concentrator. The resulting residue was reconstituted in 100 μL of 1:1 50 mM ammonium formate:MeCN and centrufuged at 13,000 rpm for 10 min in a 4°C microcentrifuge. The supernate was vialed and analyzed via LC-MS.

8 μL of each sample was analyzed on an Agilent 1200 capillary LC connected to an Agilent 6538 UHD-QToF mass spectrometer (Agilent) by injection on a Scherzo SM-18 150 x 2 mm column (Imtakt; Portland, OR) with a flow rate of 200 μL/min. For positive mode analysis, mobile phase A consisted of 0.1% formic acid in $H_2O$ and B was 0.1% formic acid in MeCN. The column was equilibrated in 100% A. The gradient was as follows: 0–5 min hold at 0% B, 0 to 100% B over 30 min, hold at 100% B for 10 min. MS data was collected from 80–1000 m/z across the entire chromatographic run, with a capillary voltage of 4000 V, a nebulizer gas flow of 11 L/min, and a pressure of 35 psig. Identical chromatographic conditions were used for metabolite identification via a targeted MS/MS analysis in which fragmentation was induced at 20 V and the product spectra collected from 50 to 1000 m/z. For negative mode analysis, mobile phase A consisted of $H_2O$ and B was 90% MeCN with 10% aqueous 50 mM ammonium formate. The column was equilibrated in 100% A. All other parameters were the same as the positive mode.

XCMS was utilized to detect and align metabolic features, providing a matrix of retention time, m/z values, and intensities for each sample (*Tautenhahn et al., 2012a*). XCMS also provides the fold change and a P-value from an univariate t-test to determine which features are dysregulated. Features were identified by exact mass (m/z), retention time, and MS/MS fragmentation by comparison against the METLIN (http://metlin.scripps.edu/) database (*Tautenhahn et al., 2012b*). The mean, minimum, and maximum ion intensities were graphed for identified metabolites using Prism.

## Statistical analyses

Comparisons between the mean variables of two groups were performed by a two-tailed Student's t-test for mRNA arrays ($RT^2$ Profiler PCR Array Data Analysis Template v3.3; Qiagen), LC/MS (XCMS; http://metlin.scripps.edu/xcms/), and other results (Excel; Microsoft). Pp <0.05 was considered to be statistically significant.

## Acknowledgements

We would like to thank Drs. Alan Bird, Mike Dorrell, and Lea Scheppke for critically evaluating the manuscript. Dr. Malcolm R Wood at the Core Microscopy Facility at TSRI provided excellent technical assistance, Dr. Yun-Zheng Le (University of Oklahoma Health Sciences Center) and Dr. Randall S Johnson (University of Cambridge) kindly provided transgenic mice. This work was supported by grants to M Friedlander from the National Eye Institute (EY-11254) and the Lowy Medical Research Institute (MacTel). T Kurihara is supported by a fellowship from the Manpei Suzuki Diabetes Foundation and The Japan Society for the Promotion of Science Postdoctoral (JSPS) Fellowships for Research Abroad. PD Westenskow is a Ruth L Kirschstein Fellow of the NIH (EY021416).

## Additional information

### Funding

| Funder | Grant reference number | Author |
| --- | --- | --- |
| National Eye Institute | EY11254 | Martin Friedlander |
| The Lowy Medical Research Institute | MacTel Project | Martin Friedlander |
| Manpei Suzuki Diabetes Foundation | Postdoctoral support | Toshihide Kurihara |
| Japan Society for the Promotion of Science Postdoctoral Fellowships for Research Abroad | Postdoctoral support | Toshihide Kurihara |

| National Eye Institute | Postdoctoral support | Peter D Westenskow |

The funders had no role in study design, data collection and interpretation, or the decision to submit the work for publication.

## Author contributions

TK, PDW, MLG, AS, SB, Conception and design, Acquisition of data, Analysis and interpretation of data, Drafting or revising the article; YU, Acquisition of data, Analysis and interpretation of data, Drafting or revising the article; EA, CW, MSHF, LPP, Acquisition of data, Drafting or revising the article; EC, Analysis and interpretation of data, Drafting or revising the article; GS, MF, Conception and design, Analysis and interpretation of data, Drafting or revising the article

## Author ORCIDs

Martin Friedlander, http://orcid.org/0000-0003-4238-9651

## Ethics

Animal experimentation: This study was performed in strict accordance with the recommendations in the Guide for the Care and Use of Laboratory Animals of the National Institutes of Health. All of the animals were handled according to approved institutional animal care and use committee (IACUC) protocols (#08-0045-3) of the Scripps Research Institute. All surgery was performed under isofluorane and/or ketamine/xylazine anesthesia, and every effort was made to minimize suffering.

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
