## [Decision Letter]

Thank you for submitting your manuscript "Hypoxia-induced metabolic stress in RPE cells is sufficient to induce photoreceptor degeneration" for consideration by *eLife*. Three experts reviewed your manuscript, and their assessments, together with my own, form the basis of this letter. As you will see, all of the reviewers were impressed with the importance and novelty of your work.

I am including the three reviews (lightly edited) at the end of this letter, as there are a variety of specific and useful suggestions in them.

Jeremy Nathans (Reviewing editor).

*Reviewer #1:*

This is a thorough, comprehensive, carefully done and clearly described study of the effects of transforming the metabolism of retinal pigment epithelium cells into a more glycolytic state.

The authors use genetic methods either to disrupt the choriocapillaris by blocking RPE from expressing *Vegfa* or to disrupt degradation of Hif1a by blocking RPE from expression the enzyme that ubuquitinylates it. Both transform the RPE into a metabolic state that would occur in response to hypoxia. The authors do a great job documenting the changes by characterizing gene expression and metabolite levels and by analyzing the morphological changes that occur in the structures and functions of the choroid, Bruch's membrane and in the photoreceptors. All the findings are convincing.

The new and useful information provided by these studies is that the RPE relies on mitochondrial based metabolism to support its ability to serve the retina, Bruch's membrane and the choroid. The RPE cannot do this with a glycolysis dominated metabolism. This is particularly interesting because the photoreceptors are the opposite, they rely on aerobic glycolysis for their survival and function and it has general implications for the metabolic relationships between these tissues and how important that is in a homeostasis that keeps the eye from entering diseased states.

Excellent paper.

*Reviewer #2:*

Toshihide Kurihara and colleagues investigated the consequences of hypoxia, or of a hypoxia-like response for RPE cells and the neural retina. They present a vast amount of data that strongly suggest that reduced oxygen availability causes dysfunction of the RPE resulting in an altered lipid and energy metabolism. Ultimately, these changes lead to reduced retinal function and photoreceptor degeneration. Furthermore, authors demonstrate that chronic activation of HIF2 and not of HIF1 is problematic for RPE cells.

Experiments are very well done, data are convincing and conclusions are sound. Although additional experiments are always possible, data provided convey a strong message.

I have only few specific points:

Authors use *Vegfa* cKO mice to show that real hypoxia causes similar phenotypes as observed in their *Vhl* cKO mice. The timing of phenotype appearance, however, seems different in the two mouse models. In their 2012 publication, authors show that the choriocapillaris is ablated within three days of inactivation of VEGF in the RPE. This should lead to the very fast manifestation of severe hypoxia for RPE with ensuing stabilization of HIF1A and HIF2A. Yet, authors state that the first signs of RPE hypoxia were observed only 6 months post VEGF inactivation (subsection “Choriocapillaris attenuation induced hypoxia in RPE and promotes photoreceptor degeneration”, first paragraph). Also, photoreceptor degeneration is observed only very late, at 18 months post VEGF inactivation. In their *Vhl* mice, however, phenotypes appear much earlier.

Connected to this point is the surprising finding that pimonidazole only labeled the RPE in *Vegfa* cKO mice (Figure 1). Besides cells of the RPE photoreceptors receive (most of) their oxygen from the blood in the choroid. Yet they do not seem to become hypoxic by the thinned choroid in these mice. Do intraretinal vessels compensate for the choroidal defects in these mice? Might this explain the very late manifestation of retinal degeneration in *Vegfa* cKO mice?

Whereas it is difficult to compare VEGF and *Vhl* cKO mice directly, this point might be briefly addressed in the Discussion.

*Vhl* cKO mice show strong accumulation of HIF1A and HIF2A in nuclei of RPE cells very early after *Vhl* ablation. This may influence RPE metabolism very early on. Judging from the ZO-1 staining (Figure 2), RPE cells in *Vhl* cKO seem less regularly shaped and some cells appear slightly larger. Did authors check for signs of RPE cell death? Is the RPE in old *Vhl* cKO mice healthy?

In addition, the ZO-1 staining may indicate a problem in tight junctions and thus in the outer blood-retina barrier in *Vhl* cKO mice. At the same time, RPE specific inactivation of *Vhl* causes dilation of the vessels in the choroid. Did authors consider that this may lead to leakage of blood through an impaired outer blood retina barrier to the retina and to edema formation? Might this be part of the relatively fast dropout of photoreceptor cells?

The accumulation of lipid droplets is very intriguing. A similar accumulation (mostly retinyl-ester) is observed in Rpe65 knockouts. Does *Vhl* inactivation affect RPE65 activity? Might this be a reason for the reduced retinal function?

*Reviewer #3:*

The study is well designed with strong data to support the conclusions.

It is essential to understand the underlying mechanisms of RPE abnormalities and the impact on retinal neuronal changes. This study is novel regarding the aspect of metabolic stress on RPE cells. In Figure 1–Figure 2, AMD-like phenotypes were observed in *Vegfa* cKO mice. In Figure 3–Figure 7: *Vhl* cKO mice (VEGFA induction is expected) showed AMD-like phenotypes and metabolic alterations.

Hypoxia (in particular HIF-2a stabilization) in RPE cells is sufficient to induce rapid and progressive retinal degeneration.

Hypoxia in RPE cells dramatically alters their metabolic activities. HIF-2 hyperstabilization promotes gross changes to lipid and glucose metabolism of RPE cells.

A relatively small change in glucose availability (16.8%) for photoreceptors induces profound retinal degeneration.

The effects of hypoxia in RPE cells is shown in two relevant models. One results from choriocapillaris vasoconstriction (VEGF mutant), and one directly results from *Vhl* ablation (and HIF-2 hyperstabilization).

The assertions made are supported by strong in vivo (multiple mouse lines) and in vitro data (human primary RPE cells).

The Seahorse mitochondrial fitness assays provide strong data that glycolysis and lipid-handling are actually perturbed.

Widespread changes in glucose and lipid metabolism (shown using multiple techniques) are observed in hypoxic RPE cells.

The observation that hypoxia in RPE cells alters their metabolic behaviors and promotes retinal degeneration is very relevant clinically. Developing novel HIF inhibitors could have therapeutic applications for some forms of AMD.

The manuscript could be improved by:

1) Clearly delineating the temporal manifestations (and ensuing effects on photoreceptors and RPE) from the VEGF and *Vhl* mutant mice.

2) Outline which (and why) of the two models (VEGF or *Vhl*) mutants is more physiologically relevant (particularly in the context of human AMD).

3) Describe the long-term consequences of VEGF and *Vhl* cKOs. Is neovascularization ever observed in the *Vhl* mutants?

4) Only ONL thickness values from the various controls are shown. Are any changes observed in RPE/choriocapillaris? ERG values could be included.

5) Provide more details regarding the VMD2-Cre line, especially regarding% recombination.

6) Describe the relevance of the dysregulated species of acyl-carnitines in *Vhl* mutants.

7) It seems that HIF induction (with increased VEGFA) has a great impact on metabolism. What the impact of VEGFA deficiency on metabolism? The authors may consider evaluating the key enzymatic profile in glycolysis and oxidative phosphorylation in the RPE/choroid complexes from *Vegfa* cKO mice; or use seahorse analysis to examine the OCR changes with anti-VEGF treatment in RPE cells.

8) HIF2a is suggested to play a key role in the metabolic switch. What are potential mechanisms for HIF2a regulation on the induction of glycolysis? Authors please comment.

---

## [Author Response]

*Reviewer #2: I have only few specific points: Authors use Vegfa cKO mice to show that real hypoxia causes similar phenotypes as observed in their Vhl cKO mice. The timing of phenotype appearance, however, seems different in the two mouse models. In their 2012 publication, authors show that the choriocapillaris is ablated within three days of inactivation of VEGF in the RPE. This should lead to the very fast manifestation of severe hypoxia for RPE with ensuing stabilization of HIF1A and HIF2A. Yet, authors state that the first signs of RPE hypoxia were observed only 6 months post VEGF inactivation (subsection “Choriocapillaris attenuation induced hypoxia in RPE and promotes photoreceptor degeneration”, first paragraph). Also, photoreceptor degeneration is observed only very late, at 18 months post VEGF inactivation. In their Vhl mice, however, phenotypes appear much earlier.*

We included in this revision a new figure (Figure 8) comparing the onset of the phenotype from both mouse lines. It is difficult to draw direct comparisons between the lines, but we can show when we first observed the central phenotypes using our tools.

We too were surprised at the slow disease progression in the VEGF mutants considering the severity of the vascular insult. We agree that the rod photoreceptors may obtain the minimal nourishment they require from the intraretinal vessels or perhaps even from neighboring support cells, (although cones should be able to draw from the same supplies). We included the following statement:

“[…]for reasons that are unclear, rod photoreceptors are less sensitive to oxygen and nutrient deprivation than cones are.”

*Connected to this point is the surprising finding that pimonidazole only labeled the RPE in Vegfa cKO mice (Figure 1). Besides cells of the RPE photoreceptors receive (most of) their oxygen from the blood in the choroid. Yet they do not seem to become hypoxic by the thinned choroid in these mice. Do intraretinal vessels compensate for the choroidal defects in these mice? Might this explain the very late manifestation of retinal degeneration in Vegfa cKO mice?* The late onset of hypoxia and lack of pimonidazole accumulation in photoreceptors in the *Vegfa* mutant mice surprised us as well. This is speculation, but the choroidal impairments could be compensated for by alternate vascular sources (intraretinal vasculature as the reviewer suggests), RPE, or Muller glia. But we also cannot rule out the possibility that the degree of hypoxia in the earliest stages is below the threshold of detection using our tools. Therefore, we added this disclaimer to the manuscript:

“However, we cannot exclude the possibility that low grade hypoxia may occur in RPE or other retinal cells earlier at subthreshold levels of detection.”

*Vhl cKO mice show strong accumulation of HIF1A and HIF2A in nuclei of RPE cells very early after Vhl ablation. This may influence RPE metabolism very early on. Judging from the ZO-1 staining (Figure 2), RPE cells in Vhl cKO seem less regularly shaped and some cells appear slightly larger. Did authors check for signs of RPE cell death? Is the RPE in old Vhl cKO mice healthy?*

Rapid ensuing changes are observed in RPE cells in the Vhl mutants. Hypertrophy is a central feature (quantified in Figure 4) along with massive lipid droplet accumulation and distended basal infoldings. While these manifestations occur early, the RPE are stable in this state for several weeks before more severe changes are observed. Eventually the RPE appear very unhealthy and appear to only weakly adhere to Bruch’s membrane. At late stages of photoreceptor degeneration significant changes are observed and blood vessels began to invade the outer retina. However, these changes occur at advanced stages so we cannot distinguish *Vhl* specific defects from retinal remodeling. We added Figure 5—figure supplement 2 and statement:

“In advanced stages of the phenotype, dramatic changes in RPE and the vasculature are observed consistent with retinal remodeling (Figure 5—figure supplement 2) (Marc et al., 2003).”

*In addition, the ZO-1 staining may indicate a problem in tight junctions and thus in the outer blood-retina barrier in Vhl cKO mice. At the same time, RPE specific inactivation of Vhl causes dilation of the vessels in the choroid. Did authors consider that this may lead to leakage of blood through an impaired outer blood retina barrier to the retina and to edema formation? Might this be part of the relatively fast dropout of photoreceptor cells?*

This is a very interesting question. We performed angiography at several stages of disease development and at times observed hints of leakage but nothing definitive.

*The accumulation of lipid droplets is very intriguing. A similar accumulation (mostly retinyl-ester) is observed in Rpe65 knockouts. Does Vhl inactivation affect RPE65 activity? Might this be a reason for the reduced retinal function?* Frankly, this is a question we had not considered. We did not attempt to distinguish the lipid inclusions in *Vhl* mutants from retinosomes that occur due to visual cycle dysfunction. However, we would argue that the lipid droplets observed in our study are the result of metabolic derangements and hypoxia for the following reasons: 1) Doug Vollrath’s group published a manuscript in JCI (PMC3007156) demonstrating that conditional ablation of OXPHOS in RPE cells (using *Best1-Cre;Tfam^fl/fl^* mice) also induces massive lipid droplet formation and photoreceptor degeneration; 2) HIF-2 hyperstability has been connected to OXPHOS inhibition and lipid droplet formation in other cell types.

Discussion: “Unlike retinosomes, lipid droplets that accumulate in RPE due to visual cycling defects (Imanishi, Gerke and Palczewski, 2004), the lipid inclusions observed in hypoxic RPE are likely a byproduct of OXPHOS and fatty acid oxidation dysregulation since similar inclusions (and RPE hypertrophy and photoreceptor atrophy) were observed in transgenic mice by conditionally inactivating OXPHOS in RPE (*Best1-Cre;Tfam^fl/fl^*) (Zhao et al., 2011).”

*Reviewer #3: The manuscript could be improved by:*

*1) Clearly delineating the temporal manifestations (and ensuing effects on photoreceptors and RPE) from the VEGF and Vhl mutant mice.*

Please see new Figure 8 in this revision that compares the temporal onset of the central phenotypes for both lines.

*2) Outline which (and why) of the two models (VEGF or Vhl) mutants is more physiologically relevant (particularly in the context of human AMD).*

We would argue that the VEGF phenotype is more physiologically relevant. The conditional loss of VEGF induces local and graded choriocapillaris defects that better represent conditions in AMD patients. *Vhl* cKO induces a very severe manifestation that may only be seen in patients with rare mutations leading to chronic activation of HIF signaling.

Please see the figure legend for Figure 8. “Changes in choriocapillaris density induce graded hypoxia in the *Vegfa* cKO line as they do in AMD patients, making it more physiologically relevant.”

*3) Describe the long-term consequences of VEGF and Vhl cKOs. Is neovascularization ever observed in the Vhl mutants?*

Please see our responses above to reviewer #2. Neovascularization is seen in very late disease stages. However, we cannot determine if the NV occurs due to chronic VEGF upregulation or due to the secondary effects of retinal degeneration and retinal remodeling.

*4) Only ONL thickness values from the various controls are shown. Are any changes observed in RPE/choriocapillaris? ERG values could be included.*

No changes were observed in the RPE/choriocapillaris and the ERG values have now been included in Figure 5—figure supplement 1.

*5) Provide more details regarding the VMD2-Cre line, especially regarding% recombination.*

This statement was added to the Results: “Transgenic mice harboring human vitelliform macular dystrophy-2 promoter-directed cre (*VMD2-Cre*) were used to ablate *Vegfa*; severe choriocapillaris vasoconstriction is observed in adult *VMD2-Cre;Vegfa^fl/fl^*mice three days post induction (dpi) (Kurihara et al., 2012).”

Please also see Figure 1—figure supplement 1 that shows the percent recombination in *VMD-Cre* mice.

*6) Describe the relevance of the dysregulated species of acyl-carnitines in Vhl mutants.*

Discussion: “The identification of a broad class of acyl-carnitines in hypoxic RPE provides additional evidence of gross glucose metabolism and lipid-handling defects; identifying the combinations of perturbed acylcarnitines is informative for diagnosing fatty acid oxidation defects and other inborn errors of metabolism (Rinaldo, Cowan and Matern, 2008).”

*7) It seems that HIF induction (with increased VEGFA) has a great impact on metabolism. What the impact of VEGFA deficiency on metabolism? The authors may consider evaluating the key enzymatic profile in glycolysis and oxidative phosphorylation in the RPE/choroid complexes from Vegfa cKO mice; or use seahorse analysis to examine the OCR changes with anti-VEGF treatment in RPE cells.*

While *Vegfa* is significantly upregulated in *Vhl* mutant mice it is ablated in *Vegfa* mutants, but the metabolic phenotype is conserved in both lines. Therefore, *Vegfa* is unlikely to induce the metabolic defects, but we cannot rule out the possibility that the enhanced levels of *Vegfa* in the *Vhl* mutants may exacerbate the metabolic dysregulations. We performed mitochondrial profiling as suggested with Affilibercept (Eylea) and DMOG treated hRPE cells and observed no significant changes between control, DMOG treated, and DMOG/Eylea treated RPE cells (Figure 9).

Author response image 1.**DOI:**
http://dx.doi.org/10.7554/eLife.14319.028

*8) HIF2a is suggested to play a key role in the metabolic switch. What are potential mechanisms for HIF2a regulation on the induction of glycolysis? Authors please comment.*

HIF1 and HIF2 have been shown to activate over 1,000 genes in various cell types including a battery of glucose metabolism and lipid-handling genes. Intriguingly, the two isoforms activate different sets of genes at different levels depending on the cell type. While most reports implicate HIF1 in the Warburg effect, HIF2 has also been shown to be the dominant regulator of OXPHOS and fatty acid oxidation in gastric cancer and clear cell renal cell carcinoma cells. Please see this additional statement in the text:

Results: “[…]suggesting that HIF-2α, as it is in other cell-types (Qui et al., 2015; Zhao et al., 2015), is the pathological HIF isoform in hypoxic RPE.”